# PromptSFL: Improving Visual Prompt Tuning For Split Federated Learning

## Abstract

Conflict arises due to the disparity between the substantial resource demands of pre-trained models and the limited available resources of federated learning (FL) participants. Split learning presents a viable approach for adapting pre-trained models to FL, involving the allocation of a small portion of the pre-trained model to clients while deploying the remaining part on a server. Moreover, the application of Visual Prompt Tuning (VPT) to pre-trained models has shown state-of-the-art performances in parameter-efficient fine-tuning methods. However, VPT exhibits unsatisfactory performance in split federated learning (SFL) compared to its performance in centralized learning. In this paper, we first identify that VPT falls short of expectations in SFL due to the insufficient generalization capability of clients. To address this issue, we propose PromptSFL, which aligns the feature spaces of prompts between clients and the server to adapt VPT for SFL. PromptSFL transmits the final prompts in clients, termed skip prompts, to the first prompts in the server, enabling clients to extract more common features from the server. Additionally, we introduce a linear layer to map the prompts from clients to the feature space in the server during this skipping process, preventing the prompts of clients from overfitting to local datasets. Moreover, to enhance the convergence speed of SFL, PromptSFL employs an adaptive learning rate for clients. Extensive experiments demonstrate the effectiveness and efficiency of PromptSFL.

## 1 Introduction

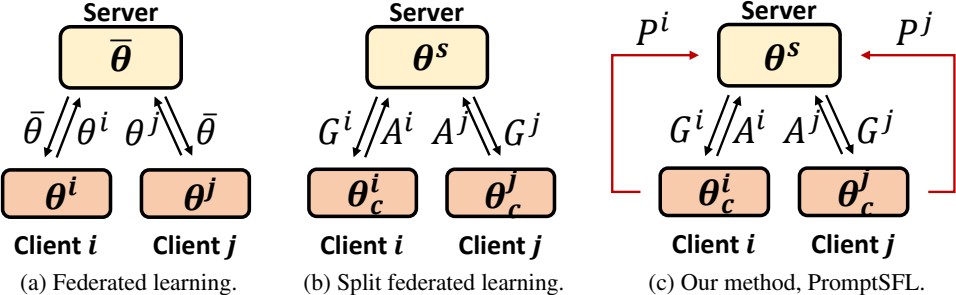

(a) Federated learning.  (b) Split federated learning.  (c) Our method, PromptSFL.

Figure 1: Differences between (a) Federated learning (FL), (b) split federated learning (SFL), and (c) our method, PromptSFL. Instead of transmitting model weights to the server, in SFL, the clients forward outputs $A$ to the server $\theta^s$, and the server sends back the gradients $G$ after completing the training process. Moreover, our method integrates the client prompts $P$ to the server.

The proliferation of large pre-trained models has sparked growing research interest in deploying and fine-tuning these models within Federated Learning (FL) (Yang et al., 2023; Chen et al., 2024; Lin et al., 2023; Qiu et al., 2024). This interest is supported by the capability of FL to leverage the advanced capabilities of pre-trained models while simultaneously protecting user data privacy (Huang et al., 2021; Li et al., 2022b; Lyu et al., 2022). However, a fundamental conflict arises between pre-trained models and FL, attributed to the extensive computational resources required for training and deploying such models, including significant memory and high-performance GPUs, which are beyond

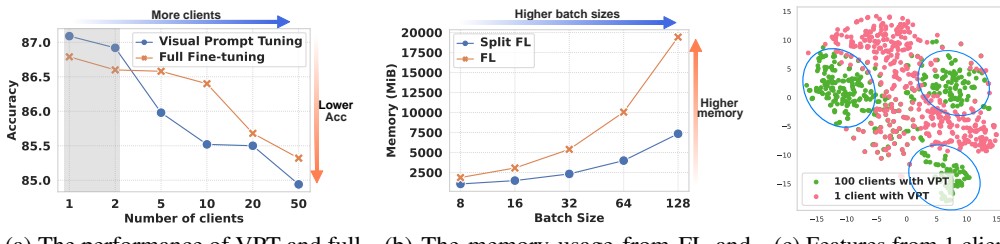

(a) The performance of VPT and full fine-tuning in SFL.

(b) The memory usage from FL and SFL in clients.

(c) Features from 1 client and 100 clients.

Figure 2: The motivations of our methods. (a) The performance of VPT and full fine-tuning (FFT) in SFL. VPT performs worse than FFT as the number of clients increases. (b) The memory usage of FL and SFL in clients. (c) T-SNE visualizations from 1 client and 100 clients in VPT-SFL. The features from 100 clients are more biased.

the reach of FL participants (Brown et al., 2020; Touvron et al., 2023). Consequently, substantial research is devoted to exploring the efficient methods for deploying and fine-tuning pre-trained models within FL environments (Zhang et al., 2024; Chen et al., 2023a; Deng et al., 2024; Yang et al., 2023; Zhang et al., 2023).

Split Federated Learning (SFL) (Thapa et al., 2022) is an efficient and effective method for deploying pre-trained models in FL, as shown in Figure 1b. In SFL, pre-trained models are partitioned into two parts between clients and a server. The portion on clients is small, while the rest of the models are handled on the server (Li et al., 2022a). As depicted in Figure 2b, the memory usage on clients in SFL is significantly lower than in FL, particularly with increasing batch sizes. Moreover, instead of training entire models on clients and the server, we focus on fine-tuning the pre-trained models. Fine-tuning in SFL provides two main advantages. Firstly, it tunes only a small subset of weights rather than the entire model, enhancing training efficiency for both clients and the server (Chen et al., 2023b; Guo et al., 2023b;a). Secondly, tuning a limited set of weights preserves the generalization ability of the pre-trained model, thereby reducing the risk of overfitting in most downstream tasks (Han et al., 2024).

Furthermore, a recent prompt tuning method, Visual Prompt Tuning (VPT) (Jia et al., 2022), has drawn significant attention among parameter-efficient fine-tuning (PEFT) methods (Sohn et al., 2023; Liu et al., 2024; Khattak et al., 2023), because of its superior performance while requiring only a small portion of weights to be updated in the input spaces. VPT fine-tunes the weights in the input spaces, freezing all pre-trained weights in each transformer block, thus maintaining the generalization ability of the pre-trained models. Pre-trained models are more adaptable to diverse local datasets with varying distributions from clients in SFL. Therefore, VPT, which fine-tunes the weights in the input spaces, is more appropriate for SFL compared to other PEFT methods. Consequently, we select VPT for SFL to adapt pre-trained models. However, the results indicate that VPT performs worse than full fine-tuning (FFT) in SFL as the number of clients increases in a dataset called CUB (Wah et al., 2011), as demonstrated in Figure 2a. These findings are in contrast to the centralized training (CT), where VPT outperforms FFT (Jia et al., 2022; Han et al., 2024), as illustrated in the gray area in Figure 2a when the number of clients is sufficiently small.

In this paper, we aim to identify the reasons why VPT performs worse than FFT in SFL with a large number of clients. As depicted in Figure 2a, the performance of VPT significantly declines with an increasing number of clients. By understanding why VPT performs worse with more clients in SFL, we can address this issue to enhance the performance of VPT, thereby surpassing FFT with a large number of clients in SFL. To this end, we analyze the outputs from the clients trained by VPT-SFL with 100 clients, and compare them to outputs from a single client trained with VPT-SFL, akin to centralized training with VPT. The t-SNE results, demonstrated in Figure 2c, indicate that the features from 100 clients are more biased and fail to form a cohesive cluster. This suggests that with a larger number of clients, VPT struggles to extract general features, leading to inferior performance.

Therefore, to improve the generalization capability of clients in SFL, we propose a method called **PromptSFL**, in which clients send their prompts to a server to extract more public features in SFL, as shown in Figure 1c. Since PromptSFL is developed from VPT-SFL, we first introduce VPT-SFL. Specifically, in VPT-SFL, as presented in Figure 3b, the full model is split into two parts. The smaller part is deployed on clients, while the remaining part of the entire model is set up on the server. Both clients and the server prepend and fine-tune visual prompts in the input space. For the server, it only

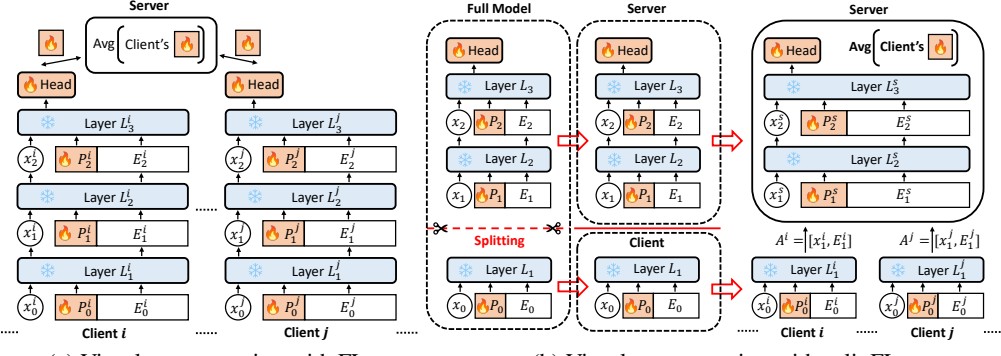

(a) Visual prompt tuning with FL.  (b) Visual prompt tuning with split FL.

Figure 3: The illustrations of the system architecture of (a) VPT-FL and (b) VPT-SFL. (a) Clients fine-tune the head layer and the prepended visual prompt tokens in the input space, then transmit fine-tuned weights to the server. (b) The full models are split into two parts. Clients send outputs $A = [x, E]$ to the server, and the server continues training. Both the server and clients fine-tune only the prepended prompt tokens and the head layer in the server.

fine-tunes visual prompts and the head layers during training. Clients send outputs $A$ to the server, and the server transmits gradients $G$ back to the clients after completing the training process, as shown in Figure 1b. Furthermore, in PromptSFL, clients forward the visual prompts from the last layer to the server, referred to as skip prompts. First, the server aligns these client prompts to the general feature space in the server using a transfer layer. Next, the server employs a learnable weight to balance the client skip prompts with the server prompts. After obtaining the mixed prompts from leveraging the skip prompts and server prompts, the server prepends these mixed prompts in the input space of the first layer and continues training. The server fine-tunes the prompts and transmits the gradients back to the clients based on the skip prompts and the client outputs $A$. Consequently, clients can extract more public knowledge from the gradients derived from the skip prompts. Moreover, we introduce a method called adaptive learning rate for clients, enhancing the convergence speed of PromptSFL. Our extensive experiments demonstrate the effectiveness of PromptSFL.

At last, our main contributions are summarized as follows,

- We first investigate the unsatisfactory performance of deploying VPT in SFL, and figure out that the client can not obtain sufficient general knowledge from VPT-SFL.
- We propose PromptSFL, which transmits the prompts from the last layer in clients to the server. The server then aligns these prompts to the general feature space, enabling clients to extract more valuable knowledge.
- To improve the convergence speed of PromptSFL, we propose a method called adaptive client learning rate, which enables the scaling of the learning rate for different clients at various phases of the training process.
- Extensive experiments on Fine-Grained Visual Classification (FGVC) tasks validate the effectiveness of PromptSFL, demonstrating its efficacy in strengthening the extraction of public information embedded in the server to clients for VPT-SFL.

## 2 PRELIMINARY

### 2.1 SPLIT FEDERATED LEARNING

Split Federated Learning (SFL) (Thapa et al., 2022) is a recently proposed approach that balances client computational resources while accommodating the pre-trained models within Federated Learning (FL). In SFL, a pivotal component is the "cut-layer" (Li et al., 2022a), which splits the full model architecture into two parts. Clients possess a part of the model including the cut-layer and preceding layers, whereas the server keeps the remaining part. Typically, the model part owned by clients is relatively smaller compared to that held by the server. As shown in Figure 1b, the original full model, denoted as $\theta$, is partitioned into $\theta = [\theta^c, \theta^s]$, where $\theta^c$ is distributed to all clients, while $\theta^s$ is retained by the server. Assuming the outputs from client $i$ are $A^i = f(\theta^i)$, $A^i$ is transmitted to the server, which continues the forward process with the $\theta^s$. Subsequently, upon computing the loss on the server, gradients $G^s = \partial L / \partial \theta^s$ are used to update $\theta^s$ on the server, and $G^s$, transmitted through $A^i$,

facilitate gradient updates on client $i$, denoted as $G^i$ in Figure 1b. Through the employment of SFL, pre-trained models can feasibly participate in the FL environment.

## 2.2 Visual Prompt Tuning

Visual Prompt Tuning (VPT) (Jia et al., 2022) is an efficient fine-tuning method that introduces prepending training prompts within the input sequence while maintaining the pre-trained model frozen. The set of prompts is defined as $P = \{P_0, P_1, ..., P_{N-1}\}$, where $P_{i-1}$ is prepended to the input space of layer $L_i$. VPT discusses two variants: VPT-Shallow and VPT-Deep. VPT-Shallow only adds prompt $P_0$ to layer $L_1$, while VPT-Deep inserts prompts across every layer. Within the context of split federated learning, we consider VPT-Deep, as prompts on deeper layers are preserved on the server, as depicted in Figure 3b, enabling the extraction of more features from different clients. In VPT-Deep, for each layer $L_i$, the forward process is defined as follows:

$$[x_i, \_, E_i] = L_i([x_{i-1}, P_{i-1}, E_{i-1}], \theta_{L_i}), i = 1, ..., N$$
$$y = Head(x_N, \theta_{Head}), \tag{1}$$

where $N$ is the number of layers, $x$ represents the **[CLS]** embeddings, and $E_{i-1}$ are the image patch embeddings serving as inputs to the layer $L_i$. In VPT-Deep, the prompts are independently injected into the input sequences of each layer. Therefore, the outputs of $L_i$ are $[x_i, \_, E_i]$, not $[x_i, P_i, E_i]$. Moreover, we color the frozen and trainable weights in Equation 1 to highlight the training weights of VPT-Deep.

## 3 Method

In this section, we introduce PromptSFL. Firstly, we present the integration of VPT-Deep with SFL, enhancing the fine-tuning capability of pre-trained models within the FL environment. Secondly, based on the observations illustrated in Figure 2c, we introduce the skip prompt mechanism, which facilitates the transmission of client prompts to the first layer on the server, thereby enabling clients to receive general knowledge from the server through skip prompts. Lastly, due to the generalized capability of the server in SFL, we propose an adaptive learning rate scheme to adjust the learning rates of clients, ensuring that client updates align more closely with the overall convergence process of the entire model.

## 3.1 VPT in Split Federated Learning

Firstly, we elaborate on the direct integration of VPT-Deep with FL (VPT-FL), enabling clients to utilize VPT-Deep for fine-tuning their pre-trained models. As presented in Figure 3a, in FL, assuming there are $N_c$ clients, each with model weights consisting of $N$ layers denoted as $\theta_{full,i}^k$, where $k = 1, ...N_c$ and $i = 1, ...N$. Employing VPT-Deep for fine-tuning a pre-trained model for client $k$ in FL is depicted as follows:

$$[x_i^k, \_, E_i^k] = L_i^k([x_{i-1}^k, P_{i-1}^k, E_{i-1}^k], \theta_{full,i}^k), i = 1, ..., N$$
$$y^k = Head^k(x_N^k, \theta_{full,H}^k), \tag{2}$$

where $\theta_{full,H}^k$ denotes the trainable weights from the linear layer $Head$. In FL, the server receives and averages the trainable prompts $P^k$ and weights $\theta_{full,H}^k$ from each client $k$.

As illustrated in Figure 3b, to extend the aforementioned method to SFL, the full model $\theta^{full}$ is first partitioned into client models $\theta^c$ and server model $\theta^s$ from $\theta^{full} = [\theta^c, \theta^s]$. If client model $k$ comprises $j$ layers, the VPT process in SFL (VPT-SFL) can be described as follows,

$$[x_i^k, \_, E_i^k] = L_i^k([x_{i-1}^k, P_{i-1}^k, E_{i-1}^k], \theta_{c,i}^k), i = 1, ..., j \tag{3}$$
$$[x_j^s, E_j^s] = [x_j^k, E_j^k] \tag{4}$$
$$[x_i^s, \_, E_i^s] = L_i^s([x_{i-1}^s, P_{i-1}^s, E_{i-1}^s], \theta_i^s), i = j + 1, ..., N \tag{5}$$
$$y^s = Head^s(x_N^s, \theta_H^s), \tag{6}$$

where Equation 4 signifies the transmission of $A^k = [x_j, E_j]$ between client $k$ and the server $s$. In SFL, both clients and the server incorporate trainable prompts $P_i$ in the input sequence, and the

server includes trainable weights $\theta_H^s$ in the $Head$ layer. During the training process, clients forward $[x_j, E_j]$ to the server. Moreover, the server computes loss based on $y^s$, updates the weights $\theta_H^s$ and prompts $P_i^s$ in the server, and backpropagates the gradients to clients according to $[x_j, E_j]$. Clients then update the prompts $P_i^k$ based on the gradients received from the server.

## 3.2 SKIP PROMPTS

However, the performance of VPT in SFL is worse than that of VPT in centralized training, because the client prompts in SFL can not extract sufficient public features, as illustrated in Figure 2. Therefore, it is crucial to provide more public knowledge to clients in SFL. In SFL, the model deployed on the server is rich in public information without introducing external data. To be specific, trainable prompts can extract public features on the server side with SFL. Thus, the sampled clients can get more features from the server prompts $P_i^s, i = j, ...N - 1$ during the federated learning process. Following this line, we propose a mechanism called skip prompts, where the prompt from the last layer in the client is transmitted to the prompt from the first layer of the server, as demonstrated in Figure 4. Specifically, the skip prompts are demonstrated as follows,

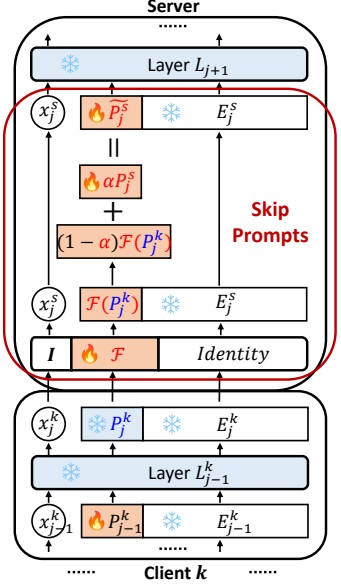

$$[x_j^s, P_j^k, E_j^s] = [x_j^k, P_j^k, E_j^k] \tag{7}$$

$$\widetilde{P_j^s} = \alpha P_j^s + (1 - \alpha)\mathcal{F}(P_j^k) \tag{8}$$

$$[x_{j+1}^s, \_, E_{j+1}^s] = L_i^s([x_j^s, \widetilde{P_j^s}, E_j^s], \theta_j^s), \tag{9}$$

where the last prompt in the client $k$ is $P_j^k$, and $P_j^k$ skips to the first layer of the server, which is inserting $P_j^k$ in the Equation 4. The final format is shown in Equation 7. Next, to align the

Figure 4: Skip prompts between the server and client $k$.

client prompt with the feature space of the server, we introduce a linear layer $\mathcal{F}$ to transfer the client prompt to the feature space of the server prompt. Moreover, the server prompt from the first layer $P_j^s$ mixes with the transferred client prompts according to a trainable weight $\alpha$, resulting in mixed prompts as shown in Equation 8. Finally, the server uses mixed prompts instead of the original prompts $P_j^s$ to train the remaining models, as shown in Equation 9. In VPT-SFL, utilizing skip prompts lets clients extract the public features from the server prompts.

## 3.3 ADAPTIVE CLIENT LEARNING RATE

However, in VPT-SFL, the gradients propagated back to clients are significantly smaller compared to those in VPT-FL, as shown in Figure 5. This phenomenon arises from the server being more generalized, as it is trained by all clients in SFL, resulting in smaller and more stable update gradients and fluctuations than the server in FL. Additionally, the paper introducing VPT (Jia et al., 2022) indicates that VPT methods require higher learning rates compared to other fine-tuning methods. Therefore, we propose a mechanism called adaptive client learning rate to enhance convergence speed and mitigate the issue of diminishing gradients for clients in VPT-SFL. Specifically, if the learning rate on the server is $\eta$, the client learning rate should be $\beta\eta$, where $\beta$ is a scaling factor that scales up the learning rate for clients to avoid gradient diminishing and enhance the convergence speed.

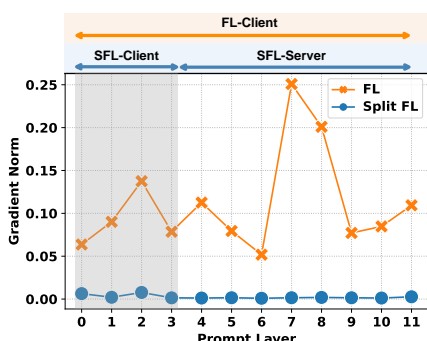

Figure 5: The norm of gradients for prompts from each layer.

Additionally, to tailor $\beta$ to various stages in the training process, we introduce an efficient method, collecting training losses before round $t_c$ and employing the moving average of training loss to

Table 1: Test accuracy of PromptSFL and baselines with Dirichlet distribution $\alpha = 1.0$. We bold the best results among all baselines for each dataset.

| Base | Methods | Fine-Tuned Params | | CUB | Flowers | Cars | Dogs | NABirds |
| --- | --- | --- | --- | --- | --- | --- | --- | --- |
| | | Client | Server | 1.0 | 1.0 | 1.0 | 1.0 | 1.0 |
| VPT (2022) | VPT-FL | 245.9K | 0 | 86.16±1.62 | 96.53±1.27 | 70.1±2.04 | 85.32±1.94 | 82.94±1.81 |
| | VPT-SFL | 23K | 222.9K | 85.5±1.76 | 97.32±1.34 | 74.02±1.58 | 89.76±1.63 | 82.65±1.24 |
| | PromptFL (2023b) | 414.5K | 153.8K | 85.97±1.98 | 97.02±1.42 | 73.76±2.41 | 87.97±2.15 | 81.23±1.46 |
| | GatedPT (2023) | 23K | 222.9K | 85.10±2.14 | 96.80±1.55 | 72.02±2.86 | 86.24±2.41 | 80.31±2.18 |
| | FedSGPT (2024) | 245.9K | 0 | 86.01±1.87 | 97.34±1.09 | 74.17±2.35 | 89.31±1.64 | 83.33±1.77 |
| | PromptSFL | 23K | 813.5K | **87.14±1.56** | **98.13±1.32** | 75.22±2.17 | **90.17±1.60** | **83.75±1.53** |
| Common | Linear | 0 | 153.8K | 84.58±1.96 | 96.64±1.41 | 68.67±1.83 | 84.11±2.76 | 80.26±2.42 |
| | Adapter (2020) | 113K | 605.7K | 86.36±1.23 | 97.16±1.04 | 71.64±1.21 | 88.58±1.35 | 82.47±1.67 |
| | Full | 22008K | 63941K | 86.69±1.14 | 97.68±0.42 | **78.87±1.76** | 88.11±0.87 | 83.51±0.58 |

control the value of $\beta$. Specifically, at communication round $t_c$, $\beta$ for client $k$ at local training epoch $t_e$ is represented as $\beta_{t_e}^k = (\sum_{t=1}^{t_c-1} loss_t + loss_{t_e}^k)/(t_c loss_{t_e}^k)$, where $loss_t$ denotes the averaged loss for communication round $t$ from selected clients, and $loss_{t_e}^k$ is the training loss for client $k$ at local training epoch $t_e$. Moreover, to alleviate the fluctuation of $\beta$ due to the instability of $loss$, we utilize the moving average of $\beta$ to derive $\beta_{t_e}^{k,ma}$ for client $k$. Specifically, with the total local training epochs denoted as $E$ and the number of selected clients for each round as $N_s$, the $\beta_{t_c}$ at communication round $t_c$ is formulated as $\beta_{t_c} = \sum_{k=1}^{N_s} \sum_{t_e=1}^{E} \beta_{t_e}^k/(N_s E)$, and the scaling factor $\beta_{t_e}^{k,ma}$ that utilized to adjust the learning rate is defined as $\beta_{t_e}^{k,ma} = (\sum_{t=1}^{t_c-1} \beta_t + \beta_{t_e}^k)/t_c$. This method facilitates the adaptation of the learning rate across different phases of the PromptSFL training process. We will discuss more insights for the adaptive learning rate in the experimental analysis.

### 3.4 COMPARISON WITH GATED PROMPT TUNING

Our work, PromptSFL, introduces a method for deploying VPT in SFL, and enhances the performance of VPT-SFL. To enable prompts in clients to extract more knowledge, we propose a method called skip prompts, as shown in Equation 8. Clients transmit prompts from the last layer to the first layer of the server, and the server mixes the client's prompts with the server's prompts. Furthermore, to align the feature space between prompts from clients and the server, we introduce a linear layer $\mathcal{F}$ to transfer the prompts of clients to the feature space of the server.

Compared with GatedPT (Yoo et al., 2023), PromptSFL requires only prompt mixing between clients and the server. GatedPT focuses on mixing prompts between the previous layer and the current layer within each layer. Moreover, PromptSFL proposes a linear layer $\mathcal{F}$ to align the feature space of prompts between clients and the server, while GatedPT ignores the alignments of prompts from different layers. At last, GatedPT is not appropriate for FL and SFL, as we demonstrate in the experimental results section.

## 4 EXPERIMENTS

In this section, we conduct experiments to demonstrate the performance and robustness of PromptSFL and perform an ablation study to analyze each component of PromptSFL. Furthermore, we illustrate how the PromptSFL achieves improvements compared to VPT in SFL. Our source codes will be released after publishing.

### 4.1 EXPERIMENT SETUP

**Datasets and Data Distribution.** We conduct our experiments based on five different Fine-Grained Visual Classification (FGVC) datasets, which are CUB-200-2011 (Wah et al., 2011), Oxford Flowers (Nilsback and Zisserman, 2008), Standford Cars (Gebru et al., 2017), Standford Dogs (Khosla et al.,

Table 2: Test accuracy of PromptSFL and baselines with Dirichlet distribution $\alpha = 0.1$. We bold the best results among all baselines for each dataset.

| Base | Methods | Fine-Tuned Params | | CUB | Flowers | Cars | Dogs | NABirds |
|------|---------|---------|--------|-----|---------|------|------|---------|
| | | Client | Server | 0.1 | 0.1 | 0.1 | 0.1 | 0.1 |
| VPT (2022) | VPT-FL | 245.9K | 0 | 84.37±2.04 | 96.31±1.59 | 61.36±2.73 | 87.22±2.30 | 81.58±2.09 |
| | VPT-SFL | 23K | 222.9K | 84.86±1.87 | 96.39±1.23 | 61.89±2.51 | 87.35±1.78 | 82.03±1.37 |
| | PromptFL (2023b) | 414.5K | 153.8K | 85.27±2.03 | 96.15±1.28 | 62.64±2.42 | 88.42±1.86 | 79.59±2.73 |
| | GatedPT (2023) | 23K | 222.9K | 84.02±2.58 | 95.21±1.75 | 60.53±3.59 | 85.34±2.37 | 77.36±3.21 |
| | FedSGPT (2024) | 245.9K | 0 | 85.93±2.64 | 96.76±1.31 | 63.64±2.96 | 87.78±2.50 | 82.84±2.03 |
| | PromptSFL | 23K | 813.5K | **86.61±2.32** | **97.89±1.16** | 68.66±2.62 | **89.45±2.21** | **83.36±1.98** |
| Common | Linear | 0 | 153.8K | 83.91±1.47 | 95.94±0.93 | <50 | 85.06±1.34 | 75.34±1.82 |
| | Adapter (2020) | 113K | 605.7K | 86.10±1.21 | 96.93±1.14 | 70.63±1.87 | 88.53±1.42 | 82.68±1.34 |
| | Full | 22008K | 63941K | 84.63±1.37 | 96.10±0.85 | **77.46±2.01** | 87.51±1.03 | 83.26±1.29 |

2011), and NABirds (Van Horn et al., 2015). We evaluate our methods under two Non-IID settings, using a Dirichlet distribution with $\alpha = 1.0$ and $\alpha = 0.1$ for all datasets.

**Baselines.** To highlight the effectiveness of PromptSFL, we compare our methods with VPT applied in (1) FL (VPT-FL) and (2) SFL (VPT-SFL). Additionally, we consider other commonly used fine-tuning methods: (3) Full Fine-Tuning (Full), which trains all weights from the backbone and the head linear layer; (4) Adapter (Pfeiffer et al., 2020; Houlsby et al., 2019), which adds randomly initialized MLP blocks with residual connection in the transformer blocks. (5) Linear, which fine-tunes only the head linear layer. Moreover, we compare PromptSFL with three state-of-the-art fine-tuning methods, (6) PromptFL (Guo et al., 2023b), which utilizes a prompt learner for each client prompt; (7) GatedPT (Yoo et al., 2023), an improved method for VPT in centralized training; (8) FedSGPT (Deng et al., 2024), which proposes shared prompts and group prompts to improve VPT in FL. Except for VPT-FL and FedSGPT, all other baselines are deployed in the SFL. To be fair comparison, we utilize the same pre-trained model ViT-B/16 for all baselines. We report the average accuracy of three different random seeds, and all methods are run on an NVIDIA GeForce RTX 3090.

**Federated Settings.** In our SFL environment, we use ViT-B/16 pre-trained on ImageNet-1K as our pre-trained model. We split the 12 transformer layers into two parts. The client part comprises the first three layers, and the server part includes the remaining layers and the head layer. In our FL environment, we have 100 clients, with a sample ratio of 0.1 for each communication round. All methods are trained over 200 communication rounds using the SGD optimizer, and we uses the best results for each baseline. The prompt tokens are set to 10 in our experiments. We use the cosine scheduler to warm up the learning rate for all baselines.

## 4.2 EVALUATION ON DIFFERENT DATASETS.

Table 1 and Table 2 demonstrate the test accuracy across five datasets in two data distributions. PromptSFL achieves the highest performance on four of the five datasets and the second-best performance on the remaining one in both distributions. Furthermore, PromptSFL outperforms all other VPT-based baselines and delivers competitive results compared to Full Fine-Tuning. Moreover, Full requires fine-tuning all model weights (22008K) on the clients, which demands substantial memory and computational resources during training. However, PromptSFL only fine-tunes a small number of weights (23K) on the clients, which is 0.1% of the weights needed for Full, making it significantly more efficient in terms of resource usage while still maintaining high performance. Although PromptSFL needs to fine-tune more weights on the server compared to other VPT-based baselines, it does not consume any client resources and still uses far fewer resources than the Full method, balancing efficiency and performance.

## 4.3 ROBUSTNESS ANALYSIS AND ABLATION STUDY.

**Effect of Layers and Tokens.** We investigate the robustness of PromptSFL in two aspects: the effects of different numbers of client layers, as illustrated in Table 3, and the impacts of varying

Table 3: Accuracy for different numbers of client layers. We bold the best results for each setting.

| Layers | Methods | Fine-Tuned Params | | CUB | Flowers | Cars | Dogs | NABirds | Avg |
|---|---|---|---|---|---|---|---|---|---|
| | | Client | Server | 1.0 | 1.0 | 1.0 | 1.0 | 1.0 | |
| 2 layers | VPT-SFL | 15.3K | 230.6K | 86.20 | 97.63 | 73.62 | 88.26 | 81.95 | 85.49 |
| | PromptSFL | 15.3K | 821.2K | **87.37** | **97.82** | **74.44** | **89.73** | **83.01** | **86.47** |
| 3 layers | VPT-SFL | 23K | 222.9K | 85.50 | 97.32 | 74.02 | 89.76 | 82.65 | 85.75 |
| | PromptSFL | 23K | 813.5K | **87.14** | **98.13** | **75.22** | **90.17** | **83.75** | **86.85** |
| 4 layers | VPT-SFL | 30.7K | 215.2K | 85.61 | 97.52 | 64.76 | 87.88 | 81.21 | 83.40 |
| | PromptSFL | 30.7K | 805.8K | **86.71** | **98.01** | **72.05** | **89.28** | **82.43** | **85.70** |

Table 4: Accuracy for different numbers of visual prompt tokens. We bold the best results for each setting. We compute the average accuracy for 4 datasets, without Cars, in the setting of 50 tokens.

| Tokens | Methods | Fine-Tuned Params | | CUB | Flowers | Cars | Dogs | NABirds | Avg |
|---|---|---|---|---|---|---|---|---|---|
| | | Client | Server | 1.0 | 1.0 | 1.0 | 1.0 | 1.0 | |
| 10 | VPT-SFL | 23K | 222.9K | 85.50 | 97.32 | 74.02 | 89.76 | 82.65 | 85.75 |
| | PromptSFL | 23K | 813.5K | **87.14** | **98.13** | **75.22** | **90.17** | **83.75** | **86.85** |
| 20 | VPT-SFL | 46K | 292K | 85.67 | **97.35** | 48.40 | 87.02 | 81.92 | 80.07 |
| | PromptSFL | 46K | 882.6K | **86.71** | 97.07 | **73.29** | **88.98** | **83.04** | **85.82** |
| 50 | VPT-SFL | 115.2K | 499.4K | 85.53 | 96.77 | <10 | 85.46 | 78.42 | 86.54 |
| | PromptSFL | 115.2K | 1.09M | **86.80** | **97.46** | 69.27 | **87.13** | **81.26** | **88.16** |

visual prompt tokens, as shown in Table 4. The hyperparameters are the same as those in Table 1 and Table 2. Table 3 presents the accuracy of VPT-SFL and PromptSFL for different numbers of client layers across various datasets in FGVC tasks. The results highlight the performance of PromptSFL as the number of client layers changes. Table 4 demonstrates the accuracy of VPT-SFL and PromptSFL for different numbers of visual prompt tokens. PromptSFL consistently outperforms VPT-SFL in this robustness analysis. Both examinations indicate that PromptSFL provides significant advantages over VPT-SFL across different configurations, whether varying the number of client layers or visual prompt tokens. The consistent performance gains suggest that PromptSFL is a robust and flexible approach that adapts well to different environmental settings.

**Ablation Study.** Our ablation study includes the following eight variations, the first four variations are from the adaptive learning rate mechanism (ALR): (1) PromptSFL w/o ALR (VPT-SFL with skip prompts), which omits the adaptive learning rate for clients, (2) PromptSFL with fixed $\beta$, which $\beta = 5$ and indicates a fixed scaling up value for client learning rates, (3) PromptSFL w/o MA $\beta$, which only utilizes the $\beta_{t_e}^k$ and without using moving average for $\beta$, (4) PromptSFL with smooth $\beta$, which obtains $\beta$ from the latest three epochs instead of using the moving average. Moreover, the following four variations are focus on the components from skip prompts: (5) PromptSFL w/o skip (VPT-SFL with adaptive learning rate), which excludes the skip prompts mechanism, (6) PromptSFL with fixed $\alpha$, which $\alpha = 0.5$ and remains constant during the fine-tuning process in Equation 8, (7) PromptSFL w/o $P^s$, which is equivalent to $\alpha = 0$ in Equation 8, (8) PromptSFL w/o $\mathcal{F}$, which ignores the linear function in the skip prompt method. Table 5 presents the results of the ablation study. The results highlight the importance of each component in the PromptSFL.

### 4.4 DEEPER INSIGHTS FOR THE IMPROVEMENTS.

**T-SNE Visualizations.** The motivation behind PromptSFL is to enhance the generalization ability of clients in VPT-SFL. To evaluate this, we conduct t-SNE visualizations for PromptSFL to compare the generalization capability of clients with that of VPT-SFL, as shown in Figure 7. Figure 7a shows

Table 5: Accuracy for ablation study. ALR indicates the Adaptive Learning Rate mechanism in clients. Skip is denoted as the skip prompts mechanism. We bold the best results for each dataset.

| From | Methods | Fine-Tuned Params | | CUB | Flowers | Cars | Dogs | NABirds | Avg |
| --- | --- | --- | --- | --- | --- | --- | --- | --- | --- |
| | | Client | Server | 1.0 | 1.0 | 1.0 | 1.0 | 1.0 | |
| | VPT-SFL | 23K | 222.9K | 85.50 | 97.32 | 74.02 | 89.76 | 82.65 | 85.85 |
| ALR | (1) PromptSFL w/o $\beta$ | 23K | 813.5K | 86.32 | 97.21 | 72.47 | 90.01 | 82.79 | 85.76 |
| | (2) PromptSFL fixed $\beta$ | 23K | 813.5K | 84.56 | 96.39 | 73.57 | 88.94 | 83.21 | 85.33 |
| | (3) PromptSFL w/o MA $\beta$ | 23K | 813.5K | 86.16 | 96.43 | 73.38 | 89.42 | 82.60 | 86.05 |
| | (4) PromptSFL smooth $\beta$ | 23K | 813.5K | 86.08 | 97.72 | 74.36 | 90.13 | 82.42 | 86.14 |
| Skip | (5) PromptSFL w/o skip | 23K | 222.9K | 86.01 | 97.30 | 73.19 | 89.54 | 81.78 | 85.56 |
| | (6) PromptSFL fixed $\alpha$ | 23K | 813.5K | 86.13 | 98.12 | 74.86 | 89.77 | 83.08 | 86.40 |
| | (7) PromptSFL w/o $P^s$ | 23K | 813.5K | 85.49 | 97.98 | 73.82 | 89.60 | 83.16 | 86.01 |
| | (8) PromptSFL w/o $\mathcal{F}$ | 23K | 222.9K | 85.10 | 97.23 | 73.76 | 89.63 | 82.98 | 85.74 |
| | PromptSFL | 23K | 813.5K | **87.14** | **98.13** | **75.22** | **90.17** | **83.75** | **86.85** |

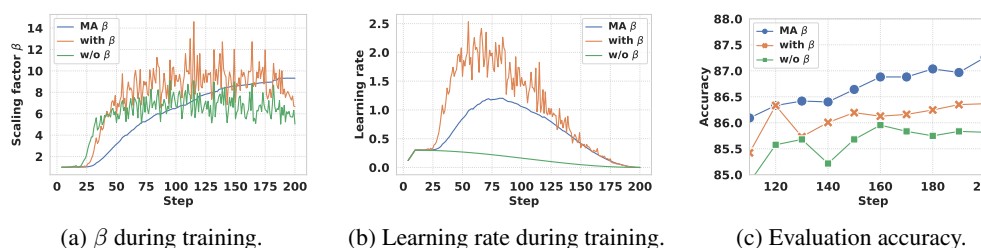

(a) $\beta$ during training.   (b) Learning rate during training.   (c) Evaluation accuracy.

Figure 6: The effects from Adaptive Learning Rate, denoted as the scaling factor $\beta$. (a) The value of scaling factor $\beta$ during training for three variations. (b) The learning rate during training. (c) Evaluation accuracy from three variations.

that the features from clients of VPT-SFL are more biased, compared to the results from 1 client with VPT. This figure indicates that VPT-SFL fails to acquire sufficient public knowledge. However, as illustrated in Figure 7b, the features from PromptSFL (blue dots) are more dispersed, similar to the features from 1 client with VPT (pink dots). This dispersion indicates that PromptSFL extracts more generalized features, leading to more generalized client models. Additionally, in Figure 7c, we evaluate the server features, which are the outputs from the last layer in the server, to deeply understand the feature space for these three methods. From figure 7c, the features from PromptSFL and 1 client with VPT are closer to each other than to those from VPT-SFL, indicating that the feature space from PromptSFL is more like the feature space from 1 client with VPT instead of VPT-SFL.

**Adaptive Learning Rates.** Figure 6 illustrates the impact of the scaling factor $\beta$. In figure 6a, two variations, with $\beta$ and w/o $\beta$, exhibit fluctuations in $\beta$ values throughout the training process, whereas MA $\beta$ remains more stable, resulting in smoother learning rates as depicted in figure 6b. Unstable and aggressive adaptive learning rates are unsuitable for PromptSFL, as evidenced in figure 6c. This is because aggressive adaptive learning rates can cause divergence in the optimized direction for clients during the training process, leading to suboptimal performance for both clients and the server.

## 5 RELATED WORK

### 5.1 SPLIT FEDERATED LEARNING

Split federated learning is first proposed by (Thapa et al., 2022), combining federated and split learning to enhance data privacy through differential privacy and PixelDP. MoCoSFL (Li et al., 2022a) is proposed as a collaborative self-supervised learning framework, leveraging split federated learning and momentum contrast (MoCo) (He et al., 2020) with a feature memory bank. FedVS (Li et al., 2023) addresses performance degradation in split vertical federated learning through a secret sharing scheme. RingSFL (Shen et al., 2023) integrates federated learning with a model split

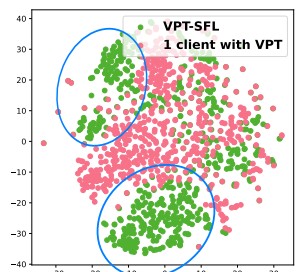 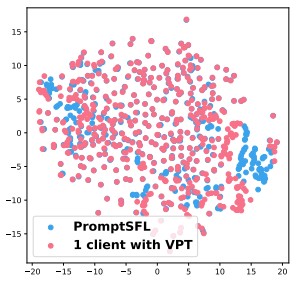 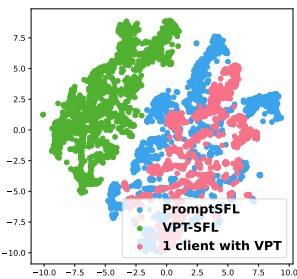

(a) Client features from VPT-SFL and 1 client with VPT.

(b) Client features from PromptSFL and 1 client with VPT.

(c) Server features from VPT-SFL, PromptSFL and 1 client.

Figure 7: T-SNE visualization of features from different methods. (a) From VPT-SFL and 1 client with VPT. (b) From PromptSFL and 1 client with VPT. (c) From VPT-SFL, PromptSFL, and 1 client with VPT. These visualizations indicate that PromptSFL can extract more general knowledge compared to VPT-SFL.

mechanism, enhancing data privacy through a specified training ring topology. FedBERT (Tian et al., 2022) applies split learning to adapt the large pre-trained model BERT to federated learning. ResSFL (Li et al., 2022b) proposes a resistant feature extractor via attacker-aware training to mitigate model inversion attacks (Fredrikson et al., 2015) on split federated learning. Moreover, some papers focusing on heterogeneity in federated learning consider splitting methods, such as HeteroFL (Diao et al., 2021), FedRolex (Alam et al., 2022), InCoFL (Chan et al., 2024), and ScaleFL (Ilhan et al., 2023). These works split models in FL based on different depths and widths to accommodate various resource requirements for different clients.

## 5.2 VISUAL PROMPT TUNING

Parameter-Efficient Fine-Tuning (PEFT) (Lester et al., 2021; Brown et al., 2020; Gao et al., 2021; Jiang et al., 2020) demonstrates its effectiveness in the NLP field, garnering attention in the CV field as well. CoOp (Zhou et al., 2022a) and CocoOp (Zhou et al., 2022b) introduce soft prompts in vision-language models. Visual Prompt Tuning (VPT) (Jia et al., 2022) is proposed to adapt the prompt tuning method into the vision transformers (ViTs) (Dosovitskiy et al., 2020) by pretending trainable prompts in the input space. GatedPT (Yoo et al., 2023) improves the performance of VPT in the ViTs pretrained with self-supervised learning through a learnable gate in each input space. $E^2$VPT (Han et al., 2023) introduces additional learnable prompts in the input space of self-attention layers. Cheng et al. (Han et al., 2024) discuss why VPT outperforms full fine-tuning in centralized training. Moreover, some studies (Deng et al., 2024; Yang et al., 2023) explore how to adapt VPT in FL. SGPT (Deng et al., 2024) introduces shared prompts and group prompts to improve VPT in FL. pFedPG (Yang et al., 2023) proposes a prompt generator to produce client-specific visual prompts, adapting VPT for personalized FL.

## 6 LIMITATIONS AND CONCLUSIONS

**Limitations.** In PromptSFL, clients need to send the prompts from the last layer to the server, which increases the communication overhead during the fine-tuning process. Additionally, because this method focuses on fine-tuning pre-trained models, privacy considerations remain a hidden limitation when the server receives fine-tuned prompts from clients. The server may potentially recover input data using methods from dataset distillation (Yu et al., 2023). These limitations require further discussion and exploration for PromptSFL.

**Conclusions.** We propose a method called PromptSFL, which aims to enhance the ability of clients to extract more general features from the server in VPT-SFL. In PromptSFL, clients send the prompts from the last layer to the server. The server transfers these prompts to the feature space of the server prompts according to a linear transfer function. Moreover, the adaptive client learning rate enhances the convergence speed of PromptSFL. Through extensive experimental evaluations, we demonstrate the effectiveness of PromptSFL and analyze the improvements for PromptSFL.

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
