# OpenReview forum: "PromptSFL: Improving Visual Prompt Tuning For Split Federated Learning"
_ICLR.cc/2025/Conference — Submitted to ICLR 2025_

### Official Review · Reviewer_pUmU · 2024-10-27

**Soundness:** 2
**Presentation:** 1
**Contribution:** 1
**Rating:** 1
**Confidence:** 5

**Summary:**

The paper proposed a solution to conduct visual prompt tuning in the split federated setting so that  the overall performance can be improved. The authors further conducted experiments to verify the effectiveness solution.

**Strengths:**

1. The authors studied an important research question relating to how we can improve training performance based on pre-trained large models in split federated learning. The scope is of a certain importance.

**Weaknesses:**

1. The reason why we need to do visual prompt fine-tuning over split federated learning is not elaborated. Since visual prompt fine-tuning performs badly over split federated learning, why do we still need to stick to it? In other words, why do authors not optimize more promising solutions over SFL. It is confused to choose an inferior solution in a particular research question and then improve it to the average bar.

2. Sending prompts to the centralized server completely breaks the privacy protocol of either split learning, federated learning, or split federated learning. In any cases, the prompt, which belongs to the user data should be sent to the server.

3. The evaluation is over image classification, which is already well-explored in the scope of federated learning and split federated learning. This back to my first point. Previous research can already achieve good performance over image classification. The authors first introduce visual prompt fine-tuning and decrease performance. And then the authors improve on this. Improvement should be based on existing state-of-the-art solutions and results. In fact, the reviewer does not think visual prompt fine-tuning should be a proper solution to choose. The reason is apparent as federated devices can suffer from heterogeneous data distribution, different centralized schemes. Visual prompt fine-tuning direct transfer knowledge between different devices can deviate the direction of correct gradients for the global model, if directly applied.

**Questions:**

1. What is the motivation for doing VPFT in SFL? Why do we need VPFT in SFL?

2. Why you selected unsupervised learning in the primary areas. How is this paper connected to unsupervised learning?

3. Is there any experiment results conducted apart from image classification?

---

### Official Review · Reviewer_hTff · 2024-10-28

**Soundness:** 2
**Presentation:** 2
**Contribution:** 2
**Rating:** 3
**Confidence:** 4

**Summary:**

This paper focuses on applying prompt tuning in split federated learning (SFL) scenarios, where prompt tuning performs worse than FFT in SFL with a large number of clients. They propose a novel method, named **PromptSFL**, to align the clients' prompts in the server, so as to improve the generalization ability. Extensive experiments indicate the better performance of **PromptSFL** compared with the state-of-the-art methods.

**Strengths:**

1. Prompt fine-tuning is an important topic for FL applications due to its efficiency in computation and communication.

2. The proposed learning framework is simple and effective.

**Weaknesses:**

1. The novelty is limited. Integrating prompt tuning with SFL is straightforward and the additional design is trivial and easy to implement.

2. The meaning of the "**skip**" is unclear. It would be better for the authors to explain in detail what is being skipped, or what is the intuition behind the design of the skip prompt.

3. Another problem is that the experimental settings for some motivation/preliminary experiments are unclear. What are the datasets and data distribution used in Figure 5? And how about the settings on data distribution, model, etc?

4. How about the performance of **PromptSFL** under different settings of clients, i.e., the number of clients ranges from 50 to 100/200, etc?

5. According to Table 1, the performance upgrade by the **PromptSFL** is marginal compared to the VPF-SFL, a more detailed analysis would be helpful to convince the reviewer.

**Questions:**

Please refer to the Weakness section.

---

### Official Review · Reviewer_nRpC · 2024-10-29

**Soundness:** 3
**Presentation:** 3
**Contribution:** 3
**Rating:** 6
**Confidence:** 4

**Summary:**

The author applies Visual Prompt Tuning (VPT) to the split federated learning task, proposing a method called PromptSFL. Based on the observed decline in VPT performance, the author suggests targeted approaches to help clients learn more general information, including the use of SKIP PROMPTS and adaptive client learning rates, along with in-depth analysis in the experimental section.

**Strengths:**

1. The application of VPT in SFL represents a relatively novel attempt within SFL, although the applicability of the proposed methods still needs verification.

2. This manuscript has a complete structure, with rich descriptions of methods and experimental setups. However, it could benefit from a deeper analysis of the challenges in fine-tuning for SFL tasks. What new difficulties arise compared to standard federated learning?

**Weaknesses:**

1. In Figure 2a, if a fixed dataset is used, the amount of data per client decreases as the number of clients increases. It raises the question of whether the lower performance of VPT compared to FFT is due to reduced data volume or if VPT is inherently less effective than FFT. Further experimentation is needed to support the conclusions drawn in Figure 2.

2. The idea of adaptively adjusting the learning rate based on loss values is interesting, but I am concerned about its suitability in the federated learning context, as it requires sharing loss values between the server and clients.

3. The claim of "improving the convergence speed" mentioned in the contributions is not clearly validated in the experiments.

**Questions:**

See the strengths 2.

---

### Official Review · Reviewer_xBhy · 2024-10-31

**Soundness:** 2
**Presentation:** 3
**Contribution:** 2
**Rating:** 5
**Confidence:** 4

**Summary:**

This work identifies the performance limitations of Visual Prompt Tuning (VPT) in Split Federated Learning (SFL) and introduces PromptSFL, which transmits client prompts to the server for alignment with general features. Prompt enhances knowledge extraction and convergence speed through an adaptive learning rate mechanism. Extensive experiments confirm PromptSFL's effectiveness in improving public information extraction for VPT-SFL on Fine-Grained Visual Classification tasks.

**Strengths:**

1. This work extends VPT to split federated learning, which is an interesting idea. PomptSFL enables clients to extract more valuable knowledge by transmitting the prompts.

2. Extensive experiments on Fine-Grained Visual Classification (FGVC) tasks show the benefits of PromptSFL compared with commonly used baselines.

**Weaknesses:**

1. This paper focuses on visual classification tasks and visual prompt tuning, which is quite limited in scope. The reviewer would like to know if and how PromptSFL could be applied to other types of tasks, such as NLP.

2. PromptSFL introduces additional computation and communication costs. A comparison of the computational overhead and communication cost of PromptSFL with other methods should be added.

3. Overall, this method seems straightforward and requires training a learnable weight and linear layer (as in Equation (8)). Is this process done on the server or on the clients?

**Questions:**

Could the authors clarify how the skip prompts are selected and how they represent the "public knowledge"?

The reviewer wonders if the value of $\alpha$ varies in experiments. As it balances the effect of client prompt and server prompt, could you provide more details on this?

Could the authors explain the baseline "Full Fine-Tuning" in more detail? Is this an FT setup where all clients tune the model? If so, why does the server also need to train the model?

A suggestion for improvement is to use the same index in the manuscript. For example, Section 2.1 uses $i$ as client index; Section 2.1 uses $L_i$ as layer index; while Section 3 uses $k$ as client index and $i$ as layer index, which is quite confusing.

---

### Meta-Review · Area_Chair_1e6x · 2024-12-17

**Metareview:**

This paper applies the Visual Prompt Tuning (VPT) to solve the split federated learning (SFL) task and propose the PromptSFL to adapt VPT for SFL. The paper initially got three negative scores and one positive scores.

The main strengths include: interesting idea, extensive experiments, and method is simple yet effective.

However, this paper also has the following weaknesses: 1) limited applications; 2) missing computational comparison; 3) unclear motivations for applying VPT to SFL and some experiments; and 4) sending prompts to server may break privacy protocol.

The authors did not submit a rebuttal. The reviewers have viewed the comments of the others and decided to keep their original ratings. Considering the drawbacks raised by the reviewers, the AC thinks this paper cannot meet the requirement of ICLR at this point and thus regrets to recommend rejection.

**Additional Comments On Reviewer Discussion:**

The authors did not submit a rebuttal. The reviewers have viewed the comments of the others' and confirmed the drawbacks raised in the first round.

---

### Decision · Program_Chairs · 2025-01-22

Reject